# The Role of Growth Hormone and Insulin Growth Factor 1 in the Development of Non-Alcoholic Steato-Hepatitis: A Systematic Review

**DOI:** 10.3390/cells12040517

**Published:** 2023-02-04

**Authors:** Luca Cristin, Amalia Montini, Alessandro Martinino, Juan Pablo Scarano Pereira, Francesco Giovinazzo, Salvatore Agnes

**Affiliations:** 1Faculty of Medicine and Surgery, University of Verona, 37134 Verona, Italy; 2Department of Immunology, University of Verona, 37134 Verona, Italy; 3Department of Surgery, University of Illinois Chicago, Chicago, IL 60607, USA; 4Faculty of Medicine, Universidad Complutense de Madrid, 28040 Madrid, Spain; 5General Surgery and Liver Transplantation Unit, Fondazione Policlinico Universitario Agostino Gemelli IRCCS, 00168 Rome, Italy

**Keywords:** growth hormone, growth hormone receptor, insulin-like growth factor-1, non-alcoholic fatty liver disease, non-alcoholic steatohepatitis

## Abstract

Diabetic and obese patients have a high prevalence of non-alcoholic fatty liver disease (NAFLD). This condition groups a spectrum of conditions varying from simple steatosis to non-alcoholic steatohepatitis (NASH), with or without fibrosis. Multiple factors are involved in the development of NAFLD. However, details about its pathogenesis and factors that promote the progression to NASH are still missing. Growth hormone (GH) and insulin-like growth factor 1 (IGF-1) regulate metabolic, immune, and hepatic stellate cell functions. Increasing evidence suggests they may have roles in the progression from NAFLD to NASH. Following the PRISMA reporting guidelines, we conducted a systematic review to evaluate all clinical and experimental studies published in the literature correlating GH and IGF-1 to inflammation and fibrosis in NAFLD and NASH. Our results showed that GH and IGF-1 have a fundamental role in the pathogenesis of NASH, acting in slightly different ways to produce a synergic effect. Indeed, GH may mediate its protective effect in the pathogenesis of NASH by regulating lipogenesis pathways, while IGF-1 has the same effect by regulating cholesterol transport. Therefore, they could be used as therapeutic strategies in preventing NAFLD progression to NASH.

## 1. Introduction

Non-alcoholic fatty liver disease (NAFLD) currently affects 20–30% of the general population. Its prevalence has increased from 15% in 2005 to 25% in 2010, and it is still expected to increase in the following years [1,2]. NAFLD is defined as a >5% fatty accumulation in the hepatocytes in the context of no excessive alcohol intake or any other hepatic pathological conditions. Ranging from simple steatosis to a more severe form of the disease known as non-alcoholic steatohepatitis (NASH), it represents a broad spectrum of pathologies [3,4]. NASH is generally characterized by four main components: liver steatosis, inflammation, hepatocellular injury, and different degrees of fibrosis [5,6,7]. These pathophysiological factors may also foster the development of cirrhosis and hepatocellular carcinoma in NASH patients, which is now considered the second most common indication for liver transplantation in the USA after alcoholic liver disease [8,9,10,11] (Figure 1).

Even though the progression of NAFLD to NASH is associated with multiple factors, two elements are generally highlighted: insulin resistance in conjunction with hepatic fat accumulation; and an increased state of oxidative stress, mitochondrial dysfunction, and inflammation [4]. All these factors have been associated to some extent with endocrine diseases; indeed, it has been reported recently that hypothyroidism, polycystic ovarian syndrome, and growth hormone (GH) deficiency may be involved in the development of NASH [2,12]. Moreover, excess lipid accumulation in the liver may be an important cause of hepatic steatosis, a condition marked by an accumulation of triglycerides in the liver. CD36 is a membrane receptor, and it is a known driver of hepato-steatosis onset, contributing significantly to its progression to NASH [13]. Indeed, CD36 takes part in free fatty acid uptake as well as triglyceride storage and secretion [14].

GH is an anterior pituitary anabolic hormone responsible for regulating lipolysis and the anti-inflammatory response [11]. It exerts its effects through insulin-like growth factor 1 (IGF-1), mainly released by the liver [4,11]. IGF-1 molecules are bound by one of the six IGF-binding proteins (IGFBP), which modulate the half-life of IGF-1 and its signaling. GH and IGF-1 are generally reduced in obesity and play a significant role in insulin sensitivity [4]. IGF-1 has also been shown to present antifibrotic properties in rodent liver disease models, including NAFLD and NASH. Thus, the state of relative GH and IGF-1 deficiency in patients with obesity may have potential implications in the development of NAFLD and NASH [11]. 

The relationship between NASH and low levels of GH and IGF-1 has not been established yet in the medical literature. In virtue of this, this systematic review aims to study the current evidence and knowledge on the pathophysiologic mechanisms of GH and IGF-1 in the development of NASH and assess the potential benefits of GH and IGF-1 therapy in this patient population.

## 2. Materials and Methods

This systematic review followed the Preferred Reporting Items for Systematic Reviews and Meta-analyses (PRISMA) reporting guidelines [15]. This study reports qualitative data, and because of the presence of different sample populations and inconsistent reporting of outcomes, a meta-analysis was not performed.

### 2.1. Searches

PubMed, Scopus, and Cochrane were searched using the following keywords: ((NASH) AND ((IGF-1) OR (GH))) to retrieve articles reporting the roles of GH and IGF-1 in the development or treatment of NASH. Results were admitted from the time of inception up to and including 17 August 2022.

### 2.2. Inclusion and Exclusion Criteria

Only published articles reporting the role of GH or IGF-1 in the development and/or treatment of NASH were included, excluding all the studies reporting GH or IGF-1’s role outside NASH. Articles describing IGF-1 and GH’s roles only in NAFLD were excluded. In human studies, only diagnoses of NASH based on biopsy were selected, while other methods like ultrasonography or imaging without biopsy confirmation were excluded. Reviews were included, while letters, editorials, conference papers, preprints, commentaries, book chapters, or any article written in a language other than English were excluded.

### 2.3. Study Selection and Data Extraction

Duplicates were removed by using the Rayyan—Intelligent Systematic Review platform [16]. Titles and abstracts of the studies were screened independently by two authors (A.Ma. and L.C.) to identify all the papers meeting the inclusion criteria. Two authors (L.C. and A.Ma.) reviewed the full texts for inclusion and data extraction. The bibliography of the included manuscripts was eventually revised to identify additional articles that could be included. Any disagreement between the screening authors was resolved through discussion with a third reviewer (F.G.).

The information extracted from the included articles was stored in a Microsoft Excel^®^ version 16.67 spreadsheet. Information related to the manuscript, such as the author’s name, journal of publication, and publication date, were collected. In each study, the number of patients, the main research question, the methodology, and the primary key findings were also gathered in the spreadsheet. At the end of the process, A.Ma. reviewed all 27 articles and rechecked that there were no mistakes in the data collection process.

## 3. Results

### 3.1. Searching Results

The study flow diagram is illustrated in Figure 2. Searches identified 104 records: 46 from PubMed, 56 from Embase, and 2 from Cochrane. A total of 14 manuscripts were duplicates and automatically excluded using the Rayyan—Intelligent Systematic Review platform [16]. A total of 90 studies were evaluated by title/abstract screening against the eligibility criteria, and 63 were excluded. Of the remaining 27 papers potentially eligible papers, after the full-text screening, 17 were included; 10 were excluded because 1 was not related to the topic, 4 were congress abstracts, 1 was in a language other than English; and 4 articles were not available. After revising the bibliography of the included articles, 4 papers were further identified. Thus, 21 cited articles were included in this systematic review that were published between 2007 and 2022 (Table 1).

### 3.2. In Vitro and In Vivo Studies (Table 2)

Fukunaga et al. described the deposition of cholesterol in hepatic tissue as a possible contribution of IGF-1 in the development of NASH [27]. This build-up depends on ABCA1 (ATP-binding cassette transporter A), a crucial regulator of reverse cholesterol transport [35]. In the study, it was seen that IGF-1 increases the expression of ABCA1 in human hepatic cells via the Phosphoinositide 3 kinase (PI3K)/Akt signaling pathway. IGF-1 administration increased ABCA1 mRNA levels in hepatic cells 3 folds compared to those in the controls (*p* < 0.05), and IGF-1 receptor blockade was found to cancel the stimulatory effect of IGF-1 on ABCA1 protein expression [27]. ABCA1 activity decreased in mice after administration of PolyEthylene Glycol (PEG), a GH receptor inhibitor. IGF-1 treatment in the PEG group considerably improved the expression and the mRNA levels of ABCA1 compared to that in the PEG without the IGF-1 group (*p* < 0.05). Administration of IGF-1 also increased HDL-cholesterol levels (by approximately 10%) and decreased AST, ALT, and TG serum levels (around 40%, 50%, and 50%, respectively) compared with those in the PEG group mice. Moreover, histological analysis revealed an improvement in hepatic fatty acid accumulation in the PEG group after IGF-1 treatment [27].

Nishizawa et al. analyzed the GH/IGF-1 axis’s role in the liver using GH-deficient rat models [4]. Serum analysis in the GH-deficient mice showed increased ALT and AST levels compared with those in the controls (mean ALT 30 IU/L vs. 12 IU/L, *p* < 0.01; mean AST 250 IU/L vs. 100 IU/L, *p* < 0,05). Histological analysis of the liver in the study group mice revealed increased TG content (9 mg/g vs. 1mg/g; *p* < 0.001) and relative fibrotic areas assessed by Masson-trichrome staining compared with those in the controls (1 vs. 3.5; *p* < 0.05). After the administration of GH, these abnormalities in the GH-deficient mice were reversed. Notably, the same effect was achieved after the administration of IGF-1 [4]. 

Nishizawa et al. also assessed the molecular mechanisms of the actions of GH and IGF-1. Quantitative real-time PCR was used to quantify the expression of three enzymes in hepatic cells: glycerol-3-phosphate acyltransferase (GPAT), glyceraldehyde-3-phosphate dehydrogenase (G3PD), and carnitine palmitoyltransferase-I (CPT-1). Relative mRNA levels of these enzymes in GH-deficient mice were significantly different compared with those in the controls (approximately GPAT was 5.5-times higher, G3PD was 3.5-times higher, and CPT-1 was 4-times lower; *p* < 0.01). Quantitative analysis also showed that the mitochondrial area in GH-deficient mice was significantly decreased compared with those in the controls (estimated 50% reduction; *p* < 0.05). Consequently, GH-deficient mice showed a significant increase in oxidative stress (*p* < 0.05). All these conditions were also reversed with the administration of GH or IGF-1 [4].

Sarmento-Cabral et al. conducted a similar study on GHR knock-down (aHepGHRkd) mice before and after the administration of IGF-1. Before or after IGF-1 administration, GHR knock-down male mice demonstrated higher liver injury marker levels than the those in the controls (*p* < 0.05). Evaluation of aHepGHRkd mice also showed increased TG levels, hepatic fatty acids, and de novo lipogenesis (DNL) compared with those in the controls. Surprisingly, even though reconstitution of IGF-1 in male GHR knock-down mice significantly improved metabolic function compared to that in the controls (*p* < 0.05), it did not affect circulating or hepatic TG levels, de novo lipogenesis (DNL), ALT levels, the degree of steatosis, ballooning, or the number of inflammatory foci [31]. 

Hosui et al. analyzed the role of signal transducer and activator of transcription 5 (STAT5) in hepatic lipid metabolism. GH binds to its receptor (GHR), which activates JAK2, which in turn phosphorylates STAT5 [36] (p. 5). Phosphorylated STAT5 translocates to the nucleus, binds to GAS motifs, and induces IGF-1 gene expression [37,38]. STAT5 action on the lipid metabolism is partially mediated by CD36, a transporter of FFA that has a pivotal role in NAFLD due to the enhancement of FFA uptake and liver steatosis [39] (p. 36). The correlation between STAT5 and CD36 was investigated through STAT5 knock-out mice (STAT5KO). Microarray analysis of liver tissue in STAT5KO mice revealed an increased expression of CD36. The qPCR analysis confirmed that CD36 expression was elevated more than 16-fold in STAT5KO mice compared with those in wild-type (WT) ones. In addition, western blot analysis revealed that CD36 protein expression levels were elevated in STAT5KO mice compared to those in the control mice [25]. Without hepatic STAT5, CD36 gene expression increased, resulting in augmented lipogenesis, fatty acid uptake, and steatosis.

In another study, Nishizawa et al. demonstrated the therapeutic effect of IGF-1 with NASH murine models. A comparison was made between NASH mice treated with and without 1 month of IGF-1. Administration of IGF-1 resulted in significantly decreased concentrations of IL-6 and fibrotic markers (relative mRNA levels of IL-6 decreased by 75%, *p* < 0.04; relative fibrotic area reduced by 70%, *p* < 0.01). Histological analysis showed a reduction in the number of cells with ballooning necrosis and in the tissue triglyceride content in mice treated with IGF-1 (ballooning necrosis cell number/HPF reduced by circa 50%, *p* < 0.03; tissue triglyceride reduced approximately by 20%, *p* < 0.05) [26]. Mitochondrial impairment and activated fibrotic factors of hepatic stellate cells (HSCs) also dramatically improved after the administration of IGF-1. The role of IGF-1 in reducing the release of HSC fibrotic factors was found to depend on p53. Indeed, HSCs lacking p53 continued to proliferate, propagating fibrosis despite IGF-1 administration [26].

**Table 2 cells-12-00517-t002:** Main findings of in vivo and in vitro studies.

Study/Year	Techniques for Analysis of Results	Main Findings	*p*-Value
**Nishizawa et al., 2012** [21]	Histological and biochemical analysis RT-PCR Immunochemical analysis	↑ Liver triglyceride content (9 mg/g vs. 1mg/g) ↑ Serum AST and ALT, steatosis, and hepatic cell injury in GH-deficient rat (SDR) (mean ALT 30 IU/L vs. 12 IU/L; mean AST 250 IU/L vs. 100 IU/L) ↓ CPT-1, ↑ expression of enzymes for triglyceride synthesis ↑ Oxidative stress markers in SDR compared with those in the control GH and/or IGF-I administration improved all these changes	Liver triglyceride content: *p* < 0.001 Serum AST: *p* < 0.05 Serum ALT: *p* < 0.01
**Nishizawa et al., 2016** [26]	Histological analysis Immunoblotting pPCR	↓ Tissue triglyceride content, ↓ cells showing ballooning necrosis, ↓ fibrosis in cells in mice NASH model treated by IGF-1(ballooning necrosis cell number/HPF reduced by circa 50%; tissue triglyceride reduced approximately by 20%) IGF-1-induced cellular senescence of HSCs IGF-1 treatment in the NASH mouse model induced a decrease in the expression of activated markers for HSCs (relative mRNA 𝛼SMA reduced by approximately 60%) p53 is necessary for the IGF-I-induced senescence in HSCs	Ballooning necrosis: cell number *p* < 0.03 Tissue triglyceride content *p* < 0.05 Expression of mRNA 𝛼SMA: *p* < 0.04
**Hosui et al., 2016** [25]	RT-PCR Western blot Immunohistochemistry and immunoblotting	↑ CD36 in STAT5KO mice compared to those in control mice (more than 16-fold) In STAT5KO mice, CD36 gene expression is increased resulting in increased lipogenesis, fatty acid uptake, and steatosis. Improvement of abnormal lipid accumulation in STAT5/CD36 double KO mice compared to those in STAT5KO mice	
**Fukunaga et al., 2018** [27]	RT-PCR Western blot Immunohistochemistry	IGF-1 increased ABCA1 mRNA expression 3 folds compared to those in the controls ABCA1 activity decreased in mice after administration of PolyEthylene Glycol (PEG), a GH receptor inhibitor. ↓ Cholesterol content in HepG2 cells treated by IGF-1 as compared to those in the control	ABCA expression *p *< 0.05 Cholesterol content: *p* < 0.05
**Sarmento-Cabral et al., 2021** [31]	Blood and hepatic lipid analysis Histological analysis	↑ TG levels, hepatic fatty acids, and de novo lipogenesis in GHR knock-down mice compared with controls Improvement in whole-body lipid oxidation, fat mass, and insulin levels in male mice treated with IGF-1. ↑ALT levels, ↑steatosis, and hepatocyte ballooning (markers of liver injury) in GHR knock-down mice even after IGF-1 administration	

ABCA1, ATP binding cassette subfamily A member 1; ALT, alanine aminotransferase; AST, aspartate aminotransferase; αSMA, alpha smooth muscle actin; BMI, body mass index; CPT1A, carnitine palmitoyltransferase 1A; GH, growth hormone; GHRH, growth hormone releasing hormone; HPF, hepatic plasma flow; HSCs, hepatic stellate cells; IGF-1, insulin-like growth factor-1; IGFBP, insulin-like growth factor-binding protein; NASH, Non-alcoholic steatohepatitis; PEG, PolyEthylene Glycol; STAT5KO, STAT5 knock-out mice; TG, triglyceride.

### 3.3. Human Studies (Table 3)

Osganian et al. analyzed the expressions of IGF-1, IGF-1R, and GH in NAFLD patients by employing Gene Expression Analysis (GEA) and Immunohistochemistry (IHC) assay [33]. IGF-1 gene expression decreased with increasing severity of NAFLD compared with those in controls for BMI and age (*p* < 0.05 in each stage of progression). IGF-1R gene and growth hormone receptor (GHR) expressions did not differ in severity in NAFLD patients [33]. 

Stanley et al. conducted a randomized, double-blinded, placebo-controlled trial that evaluated the effect of 12 months of therapy with growth hormone-releasing hormone (GHRH) analog (tesamorelin) in 61 patients affected by HIV and NAFLD. At the baseline, IGF mRNA levels were inversely related to the grade of NAFLD Activity Score (NAS) (r = −0.33, *p* = 0.04) [30]. IGF-1 bioavailability depends on IGF-1 binding proteins (IGFBPs), which bind IGFs with high affinity and modulate their actions [40]. Analyses of these proteins showed a negative correlation between IGFBP2, IGFBP4, and steatosis grade (r = −0.49, *p* < 0.02; r = −0.12, *p* < 0.02), while IGFBP6 and IGFBP7 were positively associated with steatosis (r = 0.25, *p* < 0.005; r = 0.35, *p* < 0.0001). Moreover, IGFBP6 and IGFBP7 had a strong positive relationship with NAS (r = 0.47, *p* < 0.004; r = 0.83, *p* < 0.0001), fibrosis stages (r = 0.15, *p*< 0.04; r = 0.25, *p* < 0.0001), and liver enzymes (ALT-IGFBP6 r = 0.34, *p* > 0.04; ALT-IGFBP7 r = 0.60, *p* > 0.0001; AST-IGFBP6 r = 0.33, *p* > 0.05; AST-IGFBP7 r = 0.64, *p* < 0.0001) [30]. After the treatment with tesamorelin, there were increased serum levels of IGF-1 that were associated with variations in plasma in IGFBP-1 (r = 0.31, *p* = 0.04), IGFBP-2 (r = −0.35, *p* = 0.02), and IGFBP-3 (r = 0.35, *p* = 0.02). IGFBP7 blood levels were not influenced by tesamorelin administration. Reduction of IGFBP2 was associated with a lower NAS score (r = 0.35, *p* = 0.02) and hepatocellular ballooning grade (r = 0.37, *p* = 0.02), but not with changes in the lobular inflammation grade (r = 0.17, *p* = 0.28) or steatosis grade (r = 0.17, *p* = 0.28) [30]. There were no significant associations between changes in IGFBP1, IGFBP3, or IGFBP6 and modifications in the steatosis grade or NAS.

Polyzos et al. investigated IGF-1 activity in 81 patients from Greece to evaluate its correlation with NAFLD and NASH [29]. Among the participants, 15 had biopsy-proven NAFLD and 16 had biopsy-proven NASH. These patients were compared with 50 controls (24 normal weight and 26 obese). The IGF-1 to IGFBP-3 ratio is an established marker of IGF-1 bioavailability since the main portion of serum IGF-I is bound to IGFBP-3, a growth hormone-dependent storage protein that lowers IGF-1 bioactivity [41]. In the studied population, the total IGF-1/intact IGFBP-3 ratio remained significantly lower in individuals with fibrosis than in controls with milder or no histologic lesions in the liver (*p* = 0.04). Nevertheless, after performing a binary logistic regression, the IGF-1/intact IGFBP-3 ratio did not remain robustly associated with NASH or liver fibrosis (*p* = 0.06 unadjusted and *p* = 0.08 after adjusting for BMI and age) [29]. 

In a cohort studied by Nishizawa et al., 66 Japanese subjects with adult growth hormone deficiency (AGHD) were compared with healthy age-, gender-, and BMI-matched individuals [21]. The prevalence of NAFLD in hypopituitary patients with AGHD was significantly higher than in controls (77% vs. 12%, *p* < 0.001). Among the 66 AGHD patients, 16 were treated with GH replacement therapy. Six months after GH replacement therapy, serum liver enzyme concentrations were significantly decreased (ALT: *p* < 0.001; AST: *p* < 0.005 and 𝛾-GTP: *p* < 0.05). Histological analysis showed improved steatosis (*p* = 0.04) and fibrosis (*p* = 0.04). Moreover, GH replacement therapy was related to the biochemical improvement of C-reactive protein (CRP) (from 0.6 mg/dL to 0.2 mg/dL in 6 months), hyaluronic acid (from 40 mg/dL to 20 mg/dL in 12 months; *p* < 0.05), and Type IV collagen (from approximately 4.5 mg/dL to approximately 3 mg/dL in 12 months, *p* < 0.001).

Koehler et al. presented similar results in 160 patients with obesity class III (BMI ≥ 40 kg/m^2^), wherein liver biopsies and fasting blood samples were collected. Independent from the fibrosis stage (FS), all NASH patients showed reduced GH levels compared to those in the controls (NASH with FS 0-1 patient had a median GH level of 0.10 ng/mL; NASH with FS ≥ 2 patient had a median GH level of 0.14 ng/mL; the control group had a median GH level of 0.45 ng/mL; *p* < 0.001). All patients with FS ≥ 2 had GH levels within the criteria for adult GH deficiency (<45 ng/mL) [19]. 

Ichikawa et al. investigated the relationship between hepatic fibrosis and serum levels of IGF-1, IGFBP-3, and GH in 55 patients (20 males and 35 females) with NAFLD. Patients were categorized based on the presence and type of fibrosis: pericellular fibrosis, portal fibrosis, bridging fibrosis, and ballooning. Univariate and multivariate analyses of the risk factors for advanced stages of NAFLD demonstrated a significant association between reduced IGF-1 levels and stages 2–3 of NAFLD (univariate RR = 4.643, *p* = 0.016; multivariate RR = 4.568, *p* = 0.0363). Hyaluronic acid was evaluated as a marker of hepatic fibrosis [42]. There was no correlation between GH levels and hyaluronic acid. However, IGF-1 and IGFBP-3 levels correlated negatively with the hyaluronic acid levels (r = −0.427, *p* = 0.004; r = -0.352, *p* = 0.02).

In the same study, an evaluation of fibrosis using Brunt’s classification was performed. No differences in serum IGF-1 levels were found comparing patients with hepatic pericellular fibrosis and bridging fibrosis and patients without these modifications. On the other hand, serum IGF-1 levels were significantly lower in patients with portal fibrosis (stages 1–3) than in those without it (*p* < 0.03). Patients with ballooning had lower levels of IGF-1 when compared to patients without this characteristic, even though this result was not statistically significant (*p* = 0.08). The IGF-1/IGFBP-3 ratio tended to decrease in subjects with portal fibrosis, but no statistical significance was found (*p* = 0.08).

Ichikawa et al. also found in univariate and multivariate analyses a significant correlation between GH levels and steatosis grade 2–3 evaluated with Brunt’s classification (univariate RR = 0.196, *p* = 0.0269; multivariate RR = 0.199, *p* = 0.0414). The GH/IGF-1 ratio assessment revealed a significant reduction in patients with steatosis grade 2–3 compared with those with steatosis grade 1 (*p* < 0.001). Among the different grades of steatosis, the IGF-1/IGFBP-3 ratio did not show any statistical difference [17]. 

Sumida et al. observed similar results evaluating 199 Japanese patients with biopsy-confirmed NAFLD [24]. The study group included 107 (54%) females and 130 (65%) patients with NASH. NAFLD patients demonstrated significantly lower serum levels of IGF-1 (mean value 112 ng/mL) compared to those in 2911 sex- and age-matched healthy people (mean value 121 ng/mL, *p* < 0.0001). NASH patients exhibited lower serum levels of IGF-1 (104 ng/mL) compared to those in the NAFLD patients without NASH (123 ng/mL, *p* < 0.002). Evaluation of histological factors and IGF-1 levels showed that the levels of IGF-1 increased significantly with the steatosis grade (r = 0.262, *p* < 0.001), while decreasing significantly with increasing lobular inflammation (r = −0.134, *p* < 0.001) and fibrosis (r = 0.362, *p* < 0.001). The analyses of patients with advanced fibrosis demonstrated a lower level of IGF-1 than in controls without it (80 ng/mL vs. 118 ng/mL, *p* < 0.001). The relationship between liver histological modifications and the levels of IGF-1 standard deviation score (SDS) was also investigated in the same study. The levels of IGF-1 SDS decreased significantly with increasing lobular inflammation (r = −0.272, *p* < 0.001) and fibrosis (r = −0.254, *p* < 0.001). No relationship was found between the levels of IGF-1 SDS and the steatosis grade. IGF-1 SDS levels were significantly lower in patients with advanced fibrosis than in those without it (−1.50 vs. −0.45; *p* < 0.001). Logistic regression analysis demonstrated that IGF-1 SDS was independently associated with advanced NASH (OR = 0.429, *p* < 0.0001) [24]. 

Rufinatscha et al. evaluated 29 obese women undergoing bariatric surgery (14 women with simple steatosis and 15 females with NASH matched for BMI and age) by performing liver biopsy to investigate hepatic growth hormone metabolism in patients with NASH. An evaluation of hepatic GHR and IGF-1 mRNA levels was performed. GHR mRNA levels were comparable in patients with NASH and simple steatosis. At the same time, IGF-1 mRNA was significantly reduced in patients with NASH compared to those in patients with simple steatosis (approximative reduction of 66%, *p* < 0.05). Among the 15 NASH patients, IGF-1 expression was characterized by an inverse relation to the grade of inflammation, but no statistical significance was found (*p* = 0.25) [28]. 

Dichtel et al. selected 142 human patients (46% males and 54% postmenopausal females) to assess the histological severity of NAFLD and serum IGF-1 levels. Patients with lobular inflammation and hepatocyte ballooning showed lower mean serum IGF-1 levels compared to those in patients without these abnormalities (112 ± 47 ng/mL vs. 136 ± 57 ng/mL, *p* = 0.01; 115 ± 48 ng/mL vs. 135 ± 57 ng/mL, *p* = 0.05, respectively). A higher fibrosis stage was related to lower mean serum IGF-1 levels (stage 2–4 96 ± 40 ng/mL vs. stage 0–1 125 ± 51 ng/mL, *p* = 0.005). No significant correlation was found between steatosis and mean serum IGF-1 levels (steatosis absent 133 ± 56 ng/mL vs. steatosis present 118 ± 54 ng/mL, *p* = NS). Subset analysis of patients presenting with NASH demonstrated lower mean serum IGF-1 levels than in those without NASH (115 ± 8 ng vs. 137 ± 8 ng, *p* = 0.02) [11]. 

García-Galiano et al. found comparable results by analyzing 36 patients with morbid obesity (29 females and 7 males). This population was stratified according to NAS score in three groups: non-NASH (score 0–2, 14 patients), probably-NASH (score 3–4, 13 patients), and NASH (score 5–8, 9 patients). IGF-1 level assessment showed that patients with NASH had a lower IGF-1 blood level than the non- or probably-NASH patient groups (101 ng/mL vs. 157 ng/mL, *p* < 0.006). Univariate binary logistic regression showed a significant correlation between lower IGF-1 levels and the presence of NAS > 4 (NASH). For IGF-1 < 110 ng/mL, the area below the ROC curve was found to be 0.80, while sensitivity and specificity were 0.81 and 0.67, respectively. Moreover, the univariate binary logistic analysis revealed a negative statistically significant relationship between IGF-1 levels and the presence of severe steatosis. For steatosis, it was seen that for IGF-1 < 130 ng/mL, the area below the ROC curve was found to be 0.75, while sensitivity and specificity were 0.68 and 0.79, respectively. Multivariate logistic regression analysis found IGF-1 < 130 ng/mL as an independent predictor of the degree of steatosis (OR = 0.015, *p* ≤ 0.01) and IGF-1 < 110 ng/mL as an independent predictor of NASH (OR = 0.096, *p* ≤ 0.02) [18]. 

Cianfarani et al. found that younger patients showed partially different results than adults. Data obtained from 99 obese children showed no correlation between IGF-1 levels and fibrosis [22]. IGF-1 SDS levels were inversely related to the grade of liver inflammation (r = −0.24, *p* < 0.02), steatosis (r = −0.37, *p* < 0.002), ballooning (r = −0.47, *p* < 0.001), and NAS (r = −0.49, *p* < 0.001). Similar results were found by analyzing the relationship between the IGF-1/IGFBP-3 ratio and liver inflammation (r = −0.32, *p* = 0.002), ballooning (r = −0.35, *p* = 0.001), and NAS (r = −0.37, *p* < 0.0001). Stepwise regression analysis showed that IGF-1 was the major and only significant predictor of both NAS (β = −0.457; *p* < 0.0001) and ballooning (β = −0.463; *p* < 0.0001; adjusted R2 = 0.20). Moreover, according to stepwise regression analysis, IGF-I/IGFBP-3 ratio was a significant predictor of liver inflammation (β = −0.285; *p* = 0.005). In the study population, patients with NASH were compared with patients without it. Children with NASH showed significantly lower levels of IGF-1 SDS (−1.3 vs. 1.0, *p* < 0.05) and IGF-1/IGFBP-3 ratio (0.086 vs. 0.14, *p* < 0.02) and higher grades of steatosis (2.64 vs. 1.84, *p* < 0.001), lobular inflammation (1.64 vs. 1.14, *p* = 0.002), and ballooning (1.79 vs. 0.44, *p* < 0.001).

**Table 3 cells-12-00517-t003:** Main findings of human studies.

Study/Year	Techniques for Analysis of Results	Main Findings	*p*-Value
**Ichikawa et al., 2007** [17]	Clinical, laboratory and liver histology data	↓ level of GH is associated with steatosis grade 2–3 (univariate RR = 0.196; multivariate RR = 0.199) ↓ level of IGF-1 is associated with fibrosis GH/IGF-1 ratio is significantly lower in patients with steatosis grade 2–3 compared to those with grade 1 No significant difference in GH/IGF-1 ratio among different grades of steatosis	Level of GH is associated with steatosis grade 2–3 univariate: *p* = 0.0269 multivariate; *p* = 0.0414 GH/IGF-1 ratio (*p* < 0.001)
**García-Galiano et al., 2007** [18]	Clinical, biochemical, and histologic data	↓ IGF-1 levels in patients with severe steatosis compared to those in healthy subjects and morbidly obese patients ↓ Concentration of IGF-1 in blood as compared to those in non-NASH and probable-NASH group Levels of IGF-1 <130 ng/mL as well as IGF-1 < 110 ng/mL are identified as independent predictors of hepatic steatosis and degree of NASH, respectively For IGF-1 < 130 ng/mL, the area below the ROC curve was found to be 0.75, while sensitivity and specificity were 0.68 and 0.79, respectively. For IGF-1 < 110 ng/mL, the area below the ROC curve was found to be 0.80, while sensitivity and specificity were 0.81 and 0.67, respectively.	IGF-1 levels (*p* = 0.024) Concentration of IGF (*p* = 0.006)
**Koehler et al., 2012** [19]	Biochemical and histologic data	↓ GH levels in NASH patients compared to those in controls (NASH with FS 0–1 patients had a median GH level of 0.10 ng/mL; NASH with FS ≥ 2 patients had a median GH level of 0.14 ng/mL; the controls group had a median GH level of 0.45 ng/mL) A normal GH level essentially excluded the presence of NASH with advanced fibrosis.	GH level in patients and control: *p* < 0.001 *p* < 0.001
**Nishizawa et al., 2012** [21]	Biochemical and histologic data	High prevalence of NAFLD among adult hypopituitary patients with GHD as compared to that in the control group (77% vs. 12%) ↓ Fibrotic marker concentrations, (*p* = 0.04) Improvement of histological changes in patients with NASH after GH replacement therapy) Six months after GH replacement therapy, serum liver enzyme concentrations were significantly decreased.	Prevalence of NAFLD: *p* < 0.001 Fibrotic marker concentrations: (*p* = 0.04) ALT: *p* < 0.001; AST: *p* < 0.005 and 𝛾-GTP: *p* < 0.05
**Cianfarani et al., 2014** [22]	Biochemical and histologic data	IGF-1 SDS levels were inversely related to the steatosis grade (r = −0.37), ballooning (r = −0.47), and NAS (r = −0.49). IGF-I/IGFBP-3 ratio was a significant predictor of liver inflammation (β = −0.285). ↓ levels of IGF-1 SDS and IGF-1/IGFBP-3 ratio in children with NASH and higher grades of steatosis	Steatosis grade: *p* < 0.002 Ballooning: *p* < 0.001 NAS: *p* < 0.001 IGF-I/IGFBP-3 ratio: *p* = 0.005 Levels of IGF-1 SDS: (*p* < 0.05) IGF-1/IGFBP-3 ratio: (*p* < 0.02)
**Sumida et al., 2015** [24]	Clinical and histological data	Negative relationship between IGF-1 SDS and the activity of lobular inflammation (*r* = −0.134) and fibrosis (*r* = −0.362, *p* < 0.001); No relationship between the IGF-1:SDS and steatosis grade was detected	IGF-1 SDS: *p* < 0.001 Fibrosis: *p* < 0.001
**Dichtel et al., 2017** [11]	Biochemical and histologic data	↓ Serum IGF-1 levels associated with lobular inflammation and hepatocyte ballooning (112 ± 47 ng/mL vs. 136 ± 57 ng/mL; 115 ± 48 ng/mL vs. 135 ± 57 ng/mL respectively) Subset analysis of patients presenting NASH demonstrated lower mean serum IGF-1 levels compared to the respective negative controls (115 ± 8 ng vs. 137 ± 8 ng)	Serum IGF-1 levels associated with lobular inflammation (*p* = 0.01), hepatocyte ballooning, (*p* = 0.05) Serum IGF-1 levels: *p* = 0.02
**Rufinatscha et al., 2018** [28]	Biochemical and histologic data	↓ IGF-1 mRNA in patients with NASH when compared to patients with simple steatosis (approximative reduction of 66%). Among the 15 NASH patients, IGF-1 expression was characterized by an inverse relation to the grade of inflammation, but no statistical significance was found. GHR mRNA levels were comparable in patients with NASH and simple steatosis	IGF-1 mRNA: *p* < 0.05 IGF-1 expression: *p* = 0.25
**Polyzos et al., 2020** [29]	Biochemical and clinical measurements	↓ IGF-1/intact IGFBP-3 ratio in NASH patients with fibrosis as compared to in controls with milder or no histologic lesions in the liver After performing a binary logistic regression, the IGF-1/intact IGFBP-3 ratio did not remain robustly associated with NASH or liver fibrosis (unadjusted and after adjusting for BMI and age)	IGF-1/intact IGFBP-3 ratio: *p* = 0.04 IGF-1/intact IGFBP-3 after performing a binary logistic regression (*p* = 0.06 unadjusted and *p* = 0.08 after adjusting for BMI and age)
**Stanley et al., 2021** [30]	Analysis of data from a randomized clinical trial of GHRH.	↓ Hepatic IGF-1 expression in individuals with higher grades of steatosis and higher NAS scores. ↓IGFBP2 and IGFBP4 (r = −0.49; r = −0.12) and ↑ IGFBP6 and IGFBP7 (r = 0.25; r = 0.35) with increasing steatosis Reduction of IGFBP2 after tesamorelin was associated with lower NAS score (r = 0.35) and hepatocellular ballooning grade (r = 0.37), but not with changes in lobular inflammation grade (r = 0.17) GHRH increased circulating IGFBP-1 and IGFBP-3 but decreased IGFBP-2 and IGFBP-6	IGFBP2 and IGFBP4: *p* < 0.02, *p* < 0.02 IGFBP6 and IGFBP7: *p* < 0.005, *p* < 0.0001 IGFBP2 was associated with lower NAS score: (*p* = 0.02), hepatocellular ballooning grade (*p* = 0.02), but not with changes in lobular inflammation grade (*p* = 0.28)
**Osganian et al., 2022** [33]	Gene expression analysis Immunohistochemistry	No difference in IGF-1 receptor or GH receptor gene expression across worsening stages of NAFLD/NASH compared to that in control ↓ IGF-1 gene expression across disease stages No difference in GH receptor staining intensity by the severity of NAFLD compared to that in the control group	

ALT, alanine aminotransferase; AST, aspartate aminotransferase; BMI, body mass index; FS, fibrosis score; γ-GTP, gamma-glutamyl transpeptidase; GHD, growth hormone deficiency; GH, growth hormone; GHRH, growth hormone releasing hormone; IGF-1, insulin-like growth factor-1; IGFBP, insulin-like growth factor-binding protein; NAS: nursing activities score; NASH, non-alcoholic steatohepatitis; ROC, receiver operating characteristic; RR, relative risk; SDS, standard deviation score.

## 4. Discussion

To the best of our knowledge, this is the first systematic review studying the impact of GH and IGF-1 in the development of NASH. Even though the included studies presented different sample populations and methodologies, it could be inferred that the relative levels of GH, IGF-1, and its derived factors could play a role in the development of NASH. Indeed, low levels of GH and IGF-1 were generally associated with a higher degree of fibrosis, steatosis, hepatic inflammation, and hepatocellular injury, the four main components responsible for the progressive development of NASH from NAFLD [2,6] (Figure 3).

In different experimental and human studies, decreased levels of GH and IGF-1 were associated with increased severity of NAFLD and steatosis. Stanley et al. showed a negative correlation between the baseline IGF-1 mRNA and the severity of NAFLD in subjects with HIV and hepatic steatosis [30]. Similar results were found by Cianfaraini et al. in an obese pediatric population [22]. Ichikawa et al. and Osganian et al. also found a significant correlation between low levels of GH and higher grades of steatosis in NAFLD individuals [17,33]. Garcia-Galiano et al. created a diagnostic model in which IGF-1 levels below 130 ng/mL could moderately predict the presence of steatosis [18]. These results can be explained by GH and IGF-1 roles in lipid metabolism.

GH mainly regulates IGF-1 production, and both GH and IGF-1 have an anabolic effect on metabolism. They promote lipolysis and FA oxidation and enhance LDL clearance via the expression of hepatic LDL-R [20]. GH and IGF-1 can alter metabolism through the regulation of the production and action of insulin. Hyperinsulinemia, hyperglycemia, and hyperlipidemia are observed when circulating GH and IGF-1 levels decline, which may explain their association with steatosis. Another explanation for this association could be an increased expression of ABCA1, a cholesterol transporter in charge of reducing fatty accumulation in the liver. Indeed, the administration of IGF-1 in rats increased 3 folds ABCA1 mRNA compared to those in the controls (*p* < 0.05), which led to less hepatic fatty accumulation [27]. 

Besides, GH has a strong lipolytic effect, preferentially on visceral adipose tissue, with a lesser effect on subcutaneous adipose tissue. Thus, a relative reduction in GH function could explain increased steatosis [24]. Furthermore, Hosui et al. showed that the absence of hepatic STAT5 in the GH signaling pathway increased CD36 expression, resulting in increased lipogenesis, fatty acid uptake, and lipotoxicity [25]. As a consequence, lipotoxic lipids induced endoplasmic reticulum (ER) stress, oxidative stress, and inflammasome activation that lead to hepatic inflammation and fibrosis (Figure 4). In addition, it has been suggested that GH may play a direct role in hepatic DNL since IGF-1 administration in GHR knock-down mice showed no improvement in TG levels, DNL, or steatosis [2,31]. This has been proven in murine models in which a GHR-JAK2-STAT5 deficient pathway induced a higher degree of steatosis, insulin resistance, and glucose intolerance [2,25] (Figure 5).

Nevertheless, Sumida et al. did not find a significant association between IGF-1 SDS and the degree of steatosis. A plausible explanation for this discrepancy could be the different ways of evaluating steatosis grade in distinct studies and the suboptimal sensitivity of ultrasound when hepatic fat infiltration is low (between 20–30%) [11,24]. Another possible reason for the different results regarding the correlation between steatosis and IGF-1 levels may be the application of IGF-1 SDS in Sumida et al.’s study. Indeed, the relationship between IGF-1 and steatosis may become less evident when IGF-1 SDS is applied [24,43]. Similarly, Dichtel et al. found no significant association between mean serum IGF-1 levels and steatosis [11]. Nevertheless, in this study, patients with steatosis still presented lower IGF-1 levels [11]. 

Apart from steatosis, different studies have associated the levels of IGF-1 and GH with the development of fibrosis. In the study conducted by Koehler et al. in an obese population, it was seen that adults with a fibrotic stage higher than 2 had GH levels within the criteria for adult GH deficiency [19]. Dichtel et al. also claimed that higher fibrosis stages were generally related to lower mean serum IGF-1 levels (stage 2–4 96 ± 40 ng/mL vs. stage 0–1 125 ± 51 ng/mL, *p* = 0.005) [11]. Polyzos et al. mentioned that the total IGF-1/intact IGFBP-3 ratio remained significantly lower in individuals with fibrosis than in controls with milder or no histologic lesions in the liver (*p* = 0.04), which may indicate the critical role IGF-1 plays in avoiding fibrosis [29]. The fibrotic liver stage also improved with the reconstitution of both GH and IGF-1. GH therapy replacement has already shown improvement in the levels of hepatic fibrosis, probably because of the reduction in the levels of CRP and significant decrease in the total amount of hepatic hyaluronic acid and Type IV collagen [21]. In murine models, it was also seen that the administration of IGF-1 resulted in a significant reduction of fibrosis and of different fibrotic markers released by HSCs [21]. 

Only the study conducted by Cianfarani et al. challenged these results mentioning that no correlation was found between IGF-1 levels and hepatic fibrosis [22]. This assumption might be understandable if we consider the sample population employed in the study, which was composed of children with obesity. Generally, children suffering from NASH present with a mild degree of fibrosis. Indeed, one study found that 95% of children with NASH presented with a low fibrosis score between F0 and F2 [22]. 

Chronic inflammation is a key aspect of the transition from NAFLD to NASH [6]. Several factors, such as diet, dysbiosis, or genetic factors, may be implicated in developing an increased proinflammatory state. Evaluation of the hormone relationship in liver inflammation demonstrated a crucial role of both GH and IGF-1 [19,22,27]. Indeed, patients with NASH were associated with lower IGF-1 and GH blood levels compared with the controls. In a subset analysis conducted on NASH patients, Rufinatscha et al. showed that IGF-1 expression was characterized by an inverse relation to the grade of inflammation. Still, no statistical significance was found due to the low sample size (*p* = 0.25) [28]. Moreover, regression analysis confirmed that IGF-1 was the major and only significant predictor of NAS and ballooning [18,22]. Furthermore, García-Galiano et al. even developed a ROC curve in which IGF-1 < 110 ng/mL could predict the diagnosis of NASH with an AUC of 0.80 and a sensitivity and specificity of 0.81 and 0.67, respectively. These pieces of evidence suggest a possible use of IGF-1 as a non-invasive molecular marker for NASH. Currently, the diagnosis of NASH is based on biopsy. Identifying non-invasive diagnostic molecular markers would be fundamental for the early detection of patients that may progress to more severe forms of non-alcoholic liver disease. Even though the ROC curve found by García-Galiano et al. is acceptable, the specificity is not very high. Marques et al. found that by combining the IGF-1 levels with ferritin and INR, the AUROC in distinguishing low/mild fibrosis reaches the outstanding level of 0.93 [44]. 

GH/IGF-1 axis dysregulation may contribute to the progression from NAFLD to NASH. Inflammation is a key aspect of NASH, and IGF-1 is related to it with a bidirectional relationship. Indeed, inflammatory cytokines such as interleukin (IL)-1β, tumor necrosis factor-α, and IL-6 can directly inhibit IGF-1 secretion from hepatocytes [45,46,47]. At the same time, IGF-1 directly modulates the expression of acute-phase reactants such as C-reactive protein (CRP) and fibrinogen. Moreover, the effect of IGF-1 on inflammation can also be explained through its role in insulin modulation. IGF-1 improves insulin sensitivity and reduces insulin resistance, a condition that precedes and contributes to adipose tissue inflammation [48,49]. This way, IGF-1 contributes to reducing the proinflammatory state produced by hyperinsulinemia. IGF-1’s role is also related to its effect on mitochondria. IGF-1 significantly improves mitochondrial morphology and reduces oxidative stress. Reactive oxygen species (ROS) are key signaling molecules stimulating inflammation by opening inter-endothelial junctions that promote inflammatory cell migration across the endothelial barrier [50]. IGF-1 action on the mitochondria reduces the production of ROS, and in this way, it reduces the progression of inflammation.

Even though the decline in GH and IGF-1 may play a potential role in the development of NASH, these hormones might also be used to design future therapeutical approaches [21,31]. That finding could enormously help physicians in treating this condition since there are no current licensed therapies for this pathology despite its high incidence. In a study on GH-deficient rats, IGF-1 and GH administration prevented mitochondrial alterations, TG accumulation, and fibrosis [21,26]. IGF-1 treatment significantly reduced inflammation, necrosis, and fibrosis markers (IL-6 concentration, the number of cells presenting ballooning necrosis, and HSC activation). However, in the same mice models, the steatosis grade was not reduced after 2 weeks of IGF-1 infusion. These results demonstrated the primary roles of IGF-1 and GH in different aspects of NASH. IGF-1, independently from GH, meliorates mitochondrial morphology and oxidative stress in the liver. On the other hand, GH has an essential and direct role in steatosis.

Comparable results were found in humans. Stanley et al. found that administering tesamorelin, a GHRH analog, in patients affected by HIV and NAFLD significantly improved the NAS. Tesamorelin’s action may be explained by its role in modulating IGFBPs. Indeed, it was associated with a reduction in IGFBP2, resulting in lower NAS scores and hepatocellular ballooning [51]. These results were consistent with a transcriptome-wide differential gene expression analysis in both murine models and adults with NAFLD [51]. IGFBP2 levels emerged significantly reduced both in the liver and in the serum of these subjects. Even though this study was conducted in HIV-infected individuals, it also raises the possibility of exploiting the therapeutical potential of GHRH analogs in the treatment of NAFLD and NASH in other populations. Stanley et al.’s results regarding IGFBP2 and IGFBP4 are consistent with their role in the pathogenesis of a variety of cancers and hepatic steatosis [30]. In addition, they exhibit mitogenic, metabolic effects on several cell types [52,53]. Even though IGFBP6 and IGFBP7 overexpression were not significantly correlated to steatosis grade or NAS by Stanley et al. their potential function in the pathological progression of NAFLD is still under investigation [30]. Interestingly, abnormal expressions of IGFBP6 and IGFBP7 were associated with cancer [54,55,56,57].

According to what was previously reported and supported by the current studies on GH therapy in NAFLD/NASH patients, GH and IGF-1 may be efficacious strategies in treating this disease. Two clinical trials show GH replacement therapy’s effect on liver steatosis in NAFLD patients. Even though the sample sizes were very low, the results seem to confirm the lipolytic effect of this hormone and the possible use of it as a therapeutic strategy. Currently, there is no licensed therapy specific to NASH. NASH patients without diabetes are treated with vitamin E at high dosages [58,59]. However, this treatment is associated with an increase in all causes of mortality and prostate cancer in predisposed patients [60]. In patients with diabetes and NASH, treatment is based on pioglitazone [61]. This drug is associated with an increased risk of weight gain, heart failure, and fractures [62,63,64]. GH or IGF-1 replacement therapy may avoid these adverse limitations. As suggested by Xue et al., 6-month replacement therapy with GH may be beneficial for liver enzymes and may also improve obesity-related other cardiovascular and metabolic complications typically associated with NAFLD and NASH [65]. In addition, GH and IGF-1’s role in the pathogenesis of NASH would make them ideal therapeutic targets. Moreover, identifying the distinct roles of certain IGFBPs in steatohepatitis progression may suggest a future use of these molecules as therapeutic targets for NASH. Currently, there are no clinical studies that evaluate the effect of direct IGFBP targeting in NASH patients, but this strategy should be taken into consideration in future trials.

The higher prevalence of NASH in adult growth hormone deficiency (AGHD) patients suggests a direct connection between NASH and GH. AGHD is a clinical syndrome characterized by increased visceral adiposity, insulin resistance, and abnormal lipid profile [34]. NASH demonstrates several aspects in common with the metabolic alterations of AGHD, and the two pathologies can be concurrent. Indeed, data obtained by liver biopsies of AGHD patients have shown a higher rate of non-alcoholic steatohepatitis (NASH), advanced fibrosis, and cirrhosis that require liver transplantation [66,67]. In AGHD subjects affected by NAFLD, GH replacement therapy significantly reduced serum liver enzyme concentrations. Moreover, GH replacement therapy also improved the histological changes in the liver concomitant with a reduction in the fibrotic marker concentrations in patients with NASH [21]. 

### Limitations and Strengths

The most significant limitation of this review is the small sample size of the clinical studies included—most of which had less than 100 patients. Indeed, to better comprehend the real efficacy of GH and IGF-1 in clinical settings, more studies with larger sample sizes need to be conducted. Secondly, the studies included have inconsistent reporting of outcome measures and essential differences in populations and study designs. At the same time, to the best of the authors’ knowledge, this is the first systematic review in the scientific literature focusing on GH and IGF-1’s roles in NASH. The aim is to provide an overview of the field, offering points for reflection that can guide future research in developing new targets for treating NASH.

## 5. Conclusions

A key issue in understanding the pathogenesis of non-alcoholic fatty liver disease (NAFLD) concerns identifying the mechanisms responsible for switching from simple steatosis to steatohepatitis (NASH). Furthermore, the diagnosis of NASH is essential because its presence has been associated with insulin resistance, dyslipidemia, and hyperglycemia. However, finding a reliable and non-invasive way to diagnose NASH remains challenging.

In conclusion, we discussed the critical role of IGF-1 and GH in the onset and progression of NASH. The studies we have analyzed suggest that enhancing hepatocyte GH and IGF-1 activity could be a therapeutic strategy to prevent NAFLD progression to NASH. The two hormones probably act in different pathways that result in synergic effects. GH could mediate its protective effect in the liver by regulating lipogenesis pathways [31], while IGF-1 regulates cholesterol transport and has a crucial role in fibrosis and alterations of mitochondrial morphology [27].

## Figures and Tables

**Figure 1 cells-12-00517-f001:**
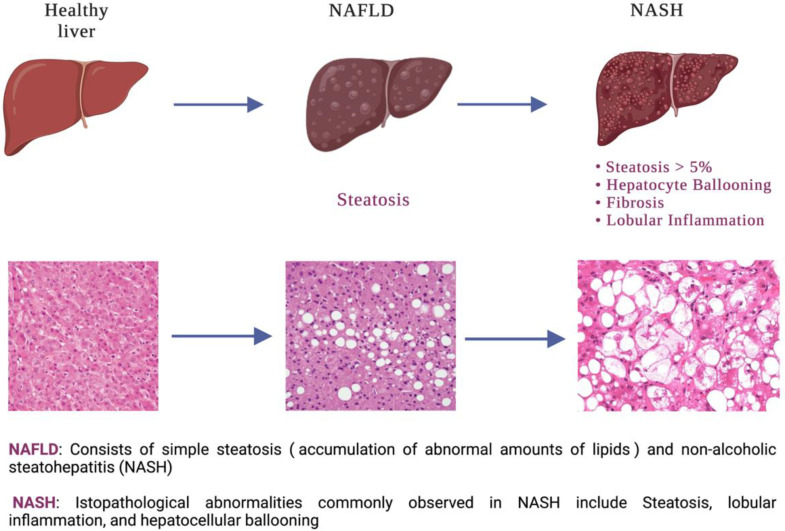
Different characteristics contributing to the transition from a healthy liver to the development of NASH.

**Figure 2 cells-12-00517-f002:**
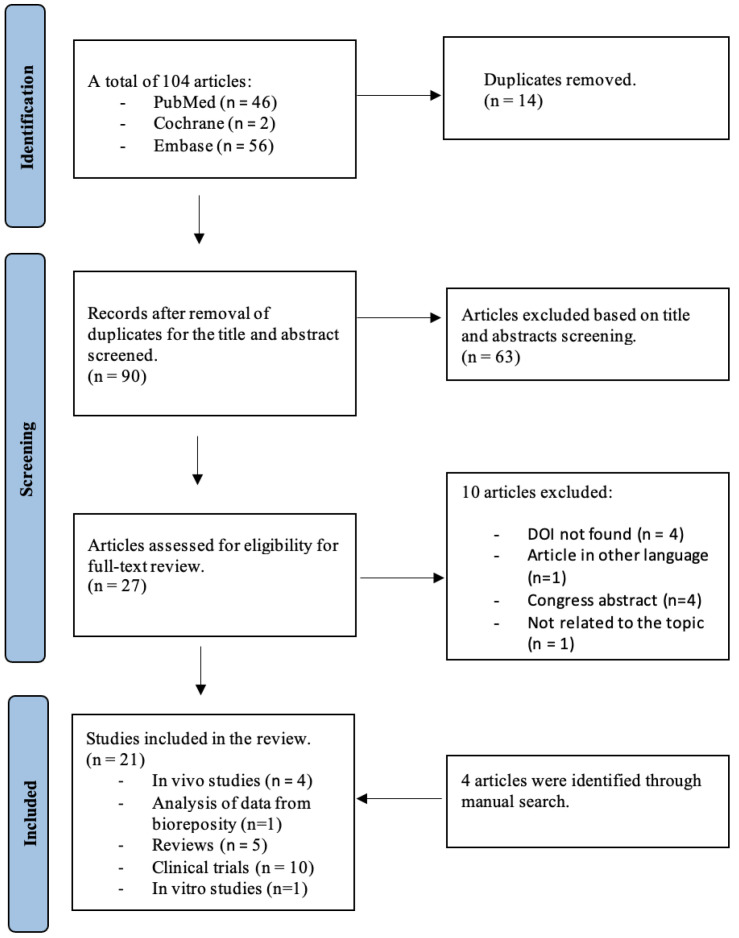
PRISMA flow diagram of the study.

**Figure 3 cells-12-00517-f003:**
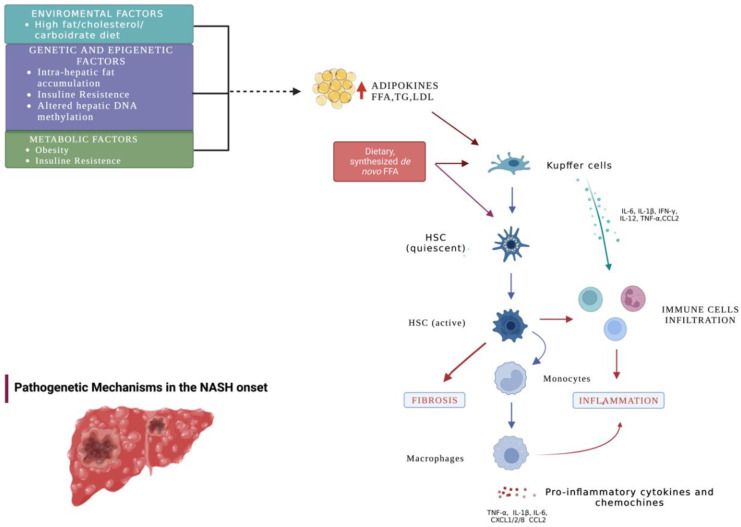
Pathogenic mechanisms in the development of NASH. Environmental, genetic, and metabolic factors are thought to be involved in the pathogenesis of NASH. Changes in the intestinal microbiome result in increased serum FFA and proinflammatory cyto/chemokine levels, believed to be the leading causes of progression to NASH. Activation of liver KCs results in the release of CCL2 TNFα, IL-1, IL-6, and other proinflammatory cytokines, leading to the recruitment of immune system cells, hepatic stellate cell activation, and macrophage-mediated inflammation.

**Figure 4 cells-12-00517-f004:**
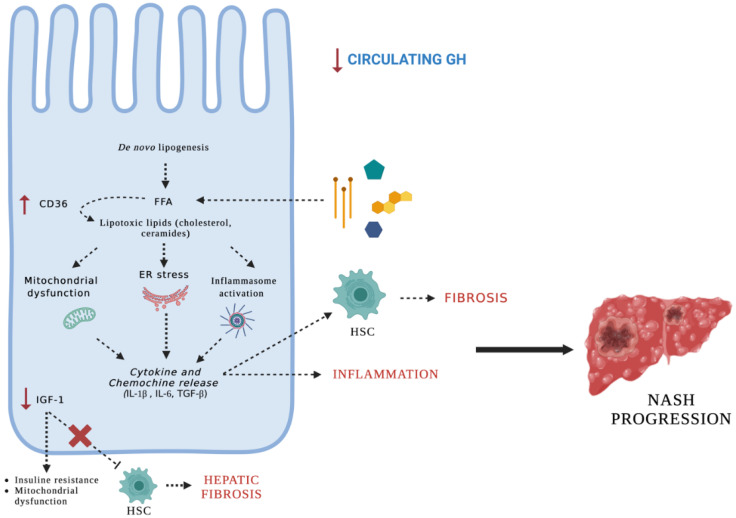
Effects of growth hormone deficiency on liver. Both hepatocyte lipotoxicity and immune-mediated inflammation are considered pivotal events in the pathogenesis of NASH. Growth hormone deficiency causes up-regulation of genes involved in de novo lipogenesis, such as CD36. Increased hepatic CD36 expression leads to higher lipogenesis, increased triglycerides (Tg) accumulation, and insulin resistance, favoring hepatic steatosis. Excess dietary fats and simple carbohydrate intake activate transcription factors that control de novo lipogenesis and mitochondrial function. Dysregulation of βoxidation induces endoplasmic reticulum (ER) stress, oxidative stress, and inflammasome activation. These events collectively lead to hepatic inflammation and fibrosis. A decrease in insulin-like growth factor-1 (IGF-1) results in insulin resistance and mitochondrial dysfunction that may further contribute to the development of NASH. Moreover, decreased IGF-1 may promote hepatic fibrosis, as IGF-1 directly inactivates hepatic stellate cells.

**Figure 5 cells-12-00517-f005:**
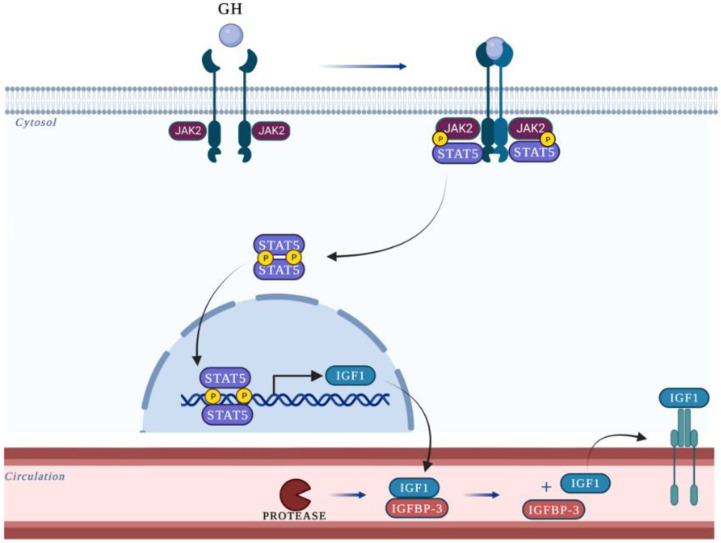
GH leads to dimerization of its receptor (GHR), promoting phosphorylation of Janus kinase (JAK) and consequent activation of STAT proteins. This event results in the dimerization and translocation of STATs to the nucleus. In the nucleus, STAT proteins are capable of binding to the IGF-1 promoter, initiating the transcription of this gene. In circulation, much of IGF-1 is bound by IGF-binding proteins (IGFBPs). Among these, IGFBP-3 is the primary carrier of IGF-1 in plasma, and it acts as a carrier protein and regulator of IGF-1 bioavailability.

**Table 1 cells-12-00517-t001:** Studies included in the systematic review assessing the relationship between NASH IGF-1 and GH.

Study/Year	Study Design	Country	N° of Patients with NAFLD/NASH	N° of Controls	M/F Ratio	Mean Population Age (±SD)	Research Question	Method of Diagnosis	Comment
**Ichikawa et al., 2007** [17]	CS	Japan	55 patients with NAFLD	-	20 males 35 females	Fibrosis stage 0–1: 47.5 ± 16.7 Fibrosis stage 2–3: 55.4 ± 17.5	Role of GH, IGF-1, IGFBP-3 in NAFLD development	Percutaneous liver biopsy	
**García-Galiano et al., 2007** [18]	CS	Spain	36 morbidly obese patients (13 with probable NASH and 9 with NASH)	12 healthy subjects	-	-	Association between the serological levels of TNF-α, IL-6, and IGF-1 with steatosis and NASH	Liver biopsy during surgery	
**Koehler et al., 2012** [19]	CS	USA	160 patients scheduled for bariatric surgery (72 with NASH and 72 with simple steatosis)	-	24 males 136 females	Normal histology: 50 ± 13.2 S.S.: 47.8 ± 11.1 NASH and FS 0-1: 46.4 ± 10 NASH and FS >2: 50.7 ± 10.7	Potential endocrine basis of steatohepatitis with advanced fibrosis in NAFLD	Liver biopsy	
**Nishizawa et al., 2012** [4]	VIVO	Japan	-	-	-	-	Effect of GH and IGF-1 administration on the liver of GH-deficient rats		5 GH-deficient rats (S.D.R.) and 5 age-matched rats as control.
**Takahashi, 2012** [20]	R	Japan	-	-	-	-	Role of GH and IGF-1 in the liver	-	
**Nishizawa et al., 2012** [21]	CS	Japan	66 patients with GHD.	1994 healthy subjects	32 males 34 females	Normal histology: 44.8 ± 16.0 NAFLD: 48.1 ± 18.0 NASH: 44.6 ± 6.4	Prevalence of NAFLD/NASH in adult hypopituitary patients with GHD	Ultrasonography and liver biopsy	
**Cianfarani et al., 2014** [22]	CS	Italy	99 obese children (14 with NASH)	-	57 males 42 females	8.73 ± 1.98	Correlate circulating levels of IGF-1, IGF2, and IGFBP3 with NASH	Liver biopsy	
**Xanthakos et al., 2014** [23]	R	USA	-	-	-	-	Correlation between abnormalities in GH axis and NASH	-	
**Sumida et al., 2015** [24]	CS	Japan	199 Japanese patients with NAFLD	2911 healthy people	92 males 107 females	NAFLD: 59 ± 10	Correlation between levels of IGF-1 SDS and histological severity of NAFLD	Liver biopsy	
**Hosui et al., 2016** [25]	IVIV	Japan	-	-	-	-	Role of STAT5 in hepatic lipid metabolism	-	STAT5KO mice and their littermates as controls.
**Nishizawa et al., 2016** [26]	IVIV	Japan	-	-	-	-	Effect of IGF-1 on NASH and cirrhotic models and the underlying mechanisms	-	NASH model, methionine-choline-deficient diet-fed db/db mice, and cirrhotic model, dimethylnitrosamine-treated mice
**Takahashi, 2017** [2]	R	Japan	-	-	-	-	Role of GH and IGF-1 in the liver	-	
**Dichtel et al., 2017** [11]	CS	USA	121 patients (80 with NASH and 41 with simple steatosis)	21 subjects	66 males 76 postmenopausal females	Normal histology: 50 ± 10 S.S.: 55 ± 8 NASH: 50 ± 11	Clarify the relationship between the histological severity of NAFLD and serum IGF-1 levels	Liver biopsy	
**Fukunaga et al., 2018** [27]	IVIV	Japan	-	-	-	-	Determine the effects of IGF-1 on ABCA1 expression in GH-deficient mice	-	3 groups (n = 5 each) of 8-week-old mice: (1) control with high-fat diet (H.F.D.) (2) H.F.D. + P.E.G. (3) H.F.D. + P.E.G. + IGF-1
**Rufinatscha et al., 2018** [28]	CS	Austria	29 obese patients (15 with NASH and 14 with simple steatosis)	-	Female population exclusively	S.S.: 37.8 ± 11.8 NASH: 45.1 ± 8.9	Role of hepatic GH signaling and its metabolic consequences in patients with NAFLD	Liver biopsy	
**Polyzos et al., 2020** [29]	CS	Greece	31 patients (16 with NASH and 15 with simple steatosis)	50 subjects (24 lean controls and 26 obese controls)	19 males 62 females	Normal histology: 56.6 ± 12.5 NASH: 56.6 ± 7.1	Evaluate hormones levels in histologically confirmed NASH patients versus S.S. patients versus controls	Liver biopsy	
**Stanley TL et al., 2021** [30]	RCT	USA	61 subjects with HIV and hepatic steatosis	-	Predominantly male	53 ± 7	Clarify the relationships between hepatic expression of IGF-1 and IGFBPs and evaluate the effect of GHRH therapy in adults with NAFLD	Liver biopsy	
**Sarmento-Cabral et al., 2021** [31]	VIVO	USA	-	-	-	-	Test the IGF-1-independent role of hepatocyte GHR signaling	-	Mice with adult-onset, hepatocyte-specific GHR knock-down (aHepGHRkd) treated with a vector expressing rat IGF-1 targeted specifically to hepatocytes
**Dichtel et al., 2022** [32]	R	USA	-	-	-	-	Roles of GH and IGF-1 in the liver and their potential application for the treatment of NASH	-	
**Osganian et al., 2022** [33]	*	USA	318 patients for the gene expression cohort and 30 for the immunohistochemistry cohort	-	-	Normal histology: 41.6 ± 11.5 S.S.: 45.2 ± 11.7 NASH F0: 43.9 ± 12.2 NASH F1-F4: 45.1 ± 12.9	IGF-1 receptor and GH receptor physiology in patients with NAFLD and NASH	Liver biopsy	
**Doycheva et al., 2022** [34]	R	USA	-	-	-	-	Pathophysiologic mechanisms of GH in NAFLD, NAFLD association with AGHD, and effect of GH treatment in patients with NAFLD	-	

CS: cross-sectional study; IVIV: in vitro and in vivo study; VIVO: in vivo study; R: review; RCT: randomized controlled trial; *: analysis of data from a NAFLD repository; AGHD: adult growth hormone deficiency; GH: growth hormone; GHD: growth hormone deficiency; GHR: growth hormone receptor; GHRH: growth hormone releasing hormone; IGF-1: insulin growth factor 1; IGF-1 SDS: insulin growth factor 1 standard deviation score; IGFBP: insulin growth factor binding proteins; NAFLD: non-alcoholic liver disease; NASH: non-alcoholic steatohepatitis; P.E.G.: PolyEthylene Glycol; S.D.R.: spontaneous dwarf rat; S.S: simple steatosis; TNF-α: tumor necrosis factor α.

## Data Availability

Data are available via email at luca98.cristin@gmail.com.

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
