# Peer review of "The Role of Growth Hormone and Insulin Growth Factor 1 in the Development of Non-Alcoholic Steato-Hepatitis: A Systematic Review"

_cells, 2023, doi:10.3390/cells12040517_

Round 1

Reviewer 1 Report

This manuscript showed that GH and IGF-1 had a preventive role in the pathogenesis of NASH using in vitro and in vivo methods. GH might mediate its protective effect in the pathogenesis of NASH by regulating lipogenesis pathways, while IGF-1 had the same effect by regulating cholesterol transport, which could be used as a therapeutic strategy to prevent NAFLD progression to NASH. Overall, the conclusion in this study should be helpful to provide appropriate strategies for NASH study and therapy. Nevertheless, several points still need to be addressed. There are some major concerns outlined below:

Major concerns.

1.     Identifying the role and mechanism of certain IGFBPs in steatohepatitis progression may help explore specific downstream targets.

2.     As described by the author, the sample size of the clinical studies is small. Furthermore, the studies had inconsistent reporting of outcome measures and differences in populations and study designs. Besides, this review is the first systematic review in the scientific literature focusing on GH and IGF-1 roles in NASH. Human research data and literature data are relatively scattered, the heterogeneity of research subjects is high and the amount of data is relatively small.

3.     In the introduction of molecular mechanisms, the relationship between IGFBPs and CD36 was not clearly explained.

4.     Whether GH and IGF can be used as non-invasive diagnostic molecular markers for non-alcoholic steatohepatitis has not been clearly described.

5.     What are the advantages of this new therapeutic strategy over traditional drug therapy should be further described.

Specific concerns.

1.     The English of the manuscript must be improved. We strongly suggest that you obtain assistance from a colleague who is well-versed in English or whose native language is English.

2.     The explanation of Figure5 in this article is too brief, and the connection between Figures 4 and 5 is not well explained.

Author Response

This manuscript showed that GH and IGF-1 had a preventive role in the pathogenesis of NASH using in vitro and in vivo methods. GH might mediate its protective effect in the pathogenesis of NASH by regulating lipogenesis pathways, while IGF-1 had the same effect by regulating cholesterol transport, which could be used as a therapeutic strategy to prevent NAFLD progression to NASH. Overall, the conclusion of this study should be helpful to provide appropriate strategies for NASH study and therapy. Nevertheless, several points still need to be addressed. There are some major concerns outlined below:

Major concerns.

  1. Identifying the role and mechanism of certain IGFBPs in steatohepatitis progression may help explore specificdownstream targets.

Thank you for your comment. We really appreciate your suggestion. We have explained the role of some IGFBPs in the discussion on lines 375-381

  1. In the introduction of molecular mechanisms, the relationship between IGFBPs and CD36 was not clearly explained.

Thank you very much for your comment. We really appreciate your suggestion; we have explained better the relationship between IGFBPs and CD36 in the introduction below figure 1.

  1. Whether GH and IGF can be used as non-invasive diagnostic molecular markers for non-alcoholic steatohepatitis has not been clearly described.

Thank you very much. We really appreciate your suggestion, and we agree that this is a very important point.  We have explained more precisely the potential role of GH and IGF-1 as non-invasive molecular markers for non-alcoholic steatohepatitis in the discussion lines 330-337

  1. What are the advantages of this new therapeutic strategy over traditional drug therapy should be further described

Thank you very much for your suggestion.  We agree that GH and IGF-1 replacement therapy advantages needed further discussion. We have explained more precisely the actual therapy of NASH and how GH and IGF-1 may have a fundamental role in future therapeutic strategies. Lines 382-396

Specific concerns.

  1. The English of the manuscript must be improved. We strongly suggest that you obtain assistance from a colleague who is well-versed in English or whose native language is English.

Thank you. A native English colleague has revised the manuscript following your suggestion.

  1. The explanation of Figure 5 in this article is too brief

Thank you very much for the suggestion. We have changed the explanation of Figure 5 adding more details.

  1. The connection between Figures 4 and 5 is not well explained

Thank you very much for your comment. We have restructured the discussion lines 259-261 to better distinguish the intracellular and molecular pathways explained in Figures 4 and 5.

Reviewer 2 Report

There are typographical errors noted. Such as the beginning of Introduction.

The present review considered a relatively low number of studies. Can the authors provide a statistical outcome based on the considered studies to show the significance of the tested hypothesis?

Authors may also provide concluding highlights of the considered studies in a tabular form.

Do the authors find some differences based on gender or age? Please clarify.

Author Response

  1. There are typographical errors noted. Such as the beginning of Introduction.

Thank you very much for your comment. We have revised the entire manuscript and deleted all the typographical errors, such as the introduction's beginning.

  1. The present review considered a relatively low number of studies. Can the authors provide a statistical outcome based on the considered studies to show the significance of the tested hypothesis?

Thank you very much for your suggestion. Unfortunately, due to the high heterogeneity of the included articles only a descriptive analysis has been conducted. However, we agree that the significance of the results presented by the studies is important. Therefore, we have created a specific column in the tables regarding the studies with a section relative to the p-value

  1. Authors may also provide concluding highlights of the considered studies in a tabular form.

Thank you very much for your comment. We have updated the tables that provide highlights of the considered studies creating a column specific to the p-values.

  1. Do the authors find some differences based on gender or age? Please clarify.

­­Thank you for your comment. Unfortunately, due to the high heterogeneity of the included articles only a descriptive analysis has been conducted, for this reason, we can not assess any difference in the total population. However, we do agree that gender and age could play a major role and for this reason, we added a new column in table 1 with the mean population age.

Reviewer 3 Report

Cristin et al present a systematic review on the association GH and insulin growth factor 1 in the pathogenesis of NAFLD/NASH. The research is within the scope of the journal. This is a well written manuscript and the methodology is well conducted. The authors have to be commended for their excellent work in summarising the current evidence on this topic which it is of great interest for future therapies as NASH becomes a global problem. I have some comments for the authors:

-In the table summarising the studies, please add each country where the study was conducted. Clearly, the pathogenesis of NAFLD/NASH and the way the study was performed could differ between countries. 

-I don't agree with the statement in the introduction "These pathophysiological factors may also foster the development of cirrhosis and hepatocellular carcinoma in NASH patients, which is now considered the second most common indication for liver transplantation in the USA after Hepatitis C". I am not sure whether the authors meant HCC as indication with background NASH vs HCV, but I think the references used by authors are old. Indications for LT are evolving after the introduction of DAA for HCV (PMID: 36096992). Alcohol and NASH are the most common indication for LT, even in US (doi:10.1001/jamanetworkopen.2019.20294). Please amend with updated references.

-Please discuss more in the depth the future perspective of your study findings with novel therapeutic approaches to prevent NAFLD/NASH progression in view of other studies trials (NCT02726542, NCT02217345  https://doi.org/10.1186/s12902-022-00967-y)

Minor comments:

The author Cianfarani has been spelled incorrectly in the discussion line 238 page 15

Figure 1: I think should be NAFLD rather than NAFL in the second image. Also, please add to the figure legend the meaning of NASH and NAFLD in extenso.

Author Response

Cristin et al present a systematic review on the association GH and insulin growth factor 1 in the pathogenesis of NAFLD/NASH. The research is within the scope of the journal. This is a well written manuscript and the methodology is well conducted. The authors have to be commended for their excellent work in summarising the current evidence on this topic which it is of great interest for future therapies as NASH becomes a global problem. I have some comments for the authors

  1. In the table summarising the studies, please add each country where the study was conducted. Clearly, the pathogenesis of NAFLD/NASH and the way the study was performed could differ between countries. 

Thank you very much for your suggestion. We agree that the country where the various studies were conducted can have fundamental importance, and for this reason, we added a new column in table 1.

  1. I don't agree with the statement in the introduction "These pathophysiological factors may also foster the development of cirrhosis and hepatocellular carcinoma in NASH patients, which is now considered the second most common indication for liver transplantation in the USA after Hepatitis C". I am not sure whether the authors meant HCC as indication with background NASH vs HCV, but I think the references used by authors are old. Indications for LT are evolving after the introduction of DAA for HCV (PMID: 36096992). Alcohol and NASH are the most common indication for LT, even in US (doi:10.1001/jamanetworkopen.2019.20294). Please amend with updated references.

Thank you for your comment. We really appreciate your suggestion; we have updated the statement in the introduction following the new references provided.

  1. Please discuss more in the depth the future perspective of your study findings with novel therapeutic approaches to prevent NAFLD/NASH progression in view of other studies trials (NCT02726542, NCT02217345 https://doi.org/10.1186/s12902-022-00967-y)
    https://pubmed.ncbi.nlm.nih.gov/36652958/

Thank you very much for your comment.  We agree that GH and IGF-1 replacement therapy future approaches needed further discussion. We have explained more precisely the actual therapy of NASH and how GH and IGF-1 may have a fundamental role in future approaches to overcoming actual problems of NASH treatment. Lines 382-396

Minor comments:

  1. The author Cianfarani has been spelled incorrectly in the discussion line 238 page 15

Thank you very much for your correction. We have promptly changed the misspelled name.

  1. Figure 1: I think should be NAFLD rather than NAFL in the second image. Also, please add to the figure legend the meaning of NASH and NAFLD in extenso.

Thank you very much for your comment. We have replaced it with a new figure with both the corrected name and the legend in extenso.

Round 2

Reviewer 1 Report

This paper uses qualitative data of different sample populations and inconsistent reporting of outcomes to show that GH and IGF-1 have a fundamental role in the pathogenesis of NASH through cross-sectional studies, in vitro and in vivo studies in human animal experiments and literature review. GH may mediate its protective effect in the pathogenesis of NASH by regulating lipogenesis pathways, while IGF-1 has the same effect by regulating cholesterol transport, which could be used as a therapeutic strategy in NAFLD progression to NASH. In conclusion, GH binds to GH receptors, which activates the JAK–STAT5 pathway increased CD36 expression induces the production of IGF-1. In circulation, much of IGF-1 is bound by IGF-binding proteins (IGFBPs),resulting in increased lipogenesis, fatty acid uptake, and steatosis. The idea of this article is enhancing hepatocyte GH and IGF-1 activity can be a therapeutic strategy to prevent NAFLD progression to NASH. The two hormones probably act in different pathways that result in synergic effects.

Major concerns.

1.     Identifying the role and mechanism of certain IGFBPs in steatohepatitis progression may explore specific targeted downstream molecules.

2.     As described by the author, the sample size of the clinical studies included is small. Secondly, the studies included have inconsistent reporting of outcome measures and essential differences in populations and study designs. Besides, this review is the first systematic review in the scientific literature focusing on GH and IGF-1 roles in NASH. Human research data and literature data are relatively scattered, the heterogeneity of research subjects is high and the amount of data is relatively small.

3.     In the introduction of molecular mechanisms, the relationship between IGFBPs and CD36 is not clearly explained.

4.     Whether GH and IGF can be used as non-invasive diagnostic molecular markers for non-alcoholic steatohepatitis has not been clearly described. This paper focuses on explaining the significance of these two mechanisms in the treatment of NASH.

5.     What are the advantages of this new therapeutic target discovery over traditional drug therapy that are not described in this review.

Specific concerns.

1.     The English of your manuscript must be improved before resubmission. We strongly suggest that you obtain assistance from a colleague who is well-versed in English or whose native language is English.

2.     The explanation of Figure5 in this article is too brief, and the connection between Figures 4 and 5 is not well explained.

The revised version has modified the picture description and writing logic according to the previous suggestions, listed the evidence and added or subtracted the problems raised before, and made some modifications to the use of English grammar and vocabulary, which is recommended to be accepted.